# Modulation Format Identification Based on Signal Constellation Diagrams and Support Vector Machine

**Zhiqi Huang** [1] , **Qi Zhang** [1,2,3,*] , **Xiangjun Xin** [4] , **Haipeng Yao** [1,2,3] , **Ran Gao** [4] , **Jinkun Jiang** [1] ,
**Feng Tian** [1,2,3] , **Bingchun Liu** [5] , **Fu Wang** [1,2,3] , **Qinghua Tian** [1,2,3] , **Yongjun Wang** [1,2,3] and **Leijing Yang** [1,2,3]

1. School of Electronic Engineering, Beijing University of Posts and Telecommunications (BUPT), Beijing 100876, China
2. Beijing Key Laboratory of Space-Ground Interconnection and Convergence, Beijing University of Posts and Telecommunications (BUPT), Beijing 100876, China
3. State Key Laboratory of Information Photonics and Optical Communications, Beijing University of Posts and Telecommunications (BUPT), Beijing 100876, China
4. The Advanced Research Institute of Multidisciplinary Science, Beijing Institute of Technology, Beijing 100081, China
5. School of Management, Tianjin University of Technology, Tianjin 300191, China
* Correspondence: zhangqi@bupt.edu.cn

**Abstract:** In coherent optical communication systems, where multiple modulation formats are mixed and variable, the correct identification of signal modulation formats provides the foundation for subsequent performance improvement using digital algorithms. A modulation format identification (MFI) scheme based on signal constellation diagrams and support vector machine (SVM) is proposed. Firstly, the signal constellation diagrams are divided by the fractal dimension of the weighted linear least squares (WLS-FD) algorithm, and the fractal dimension (FD) in each region is calculated, which is regarded as one of the image features. Then, the feature values of the image in different directions are extracted by the gray-level co-occurrence matrix (GLCM), and their mean and variance are calculated, which is regarded as another feature. Finally, the two features are input into the modulation format classifier constructed by the SVM to achieve MFI in coherent optical communication systems. To verify the feasibility and superiority of the scheme, we compare it with the MFI scheme based on higher-order statistical (HOS) features, GLCM features, and FD features, respectively. Further, we built a 30 GBaud coherent optical communication system with fiber lengths of 80 km and 120 km, where the optical signal-to-noise ratio (OSNR) ranges from 0 dB to 30 dB. The proposed MFI scheme identifies seven modulation formats: QPSK, 8QAM, 16QAM, 32QAM, 64QAM, 128QAM, and 256QAM. The results show that compared with the other three schemes, our proposed scheme has a better identification accuracy at low OSNR. In addition, the identification accuracy of this scheme can reach 100% when the OSNR $\geq$ 10 dB.

**Keywords:** signal constellation diagrams; support vector machine (SVM); coherent optical communication; modulation format identification

## 1. Introduction

With the rapid development of services such as 5G, cloud computing, high definition video, and cloud conferencing, the demand for bandwidth and spectrum utilization is increasing [1–5], and the channel capacity in optical communication systems is growing exponentially. The future fiber optic communication systems are expected to be dynamic and diverse, which can accommodate a wide variety of signals and have different modulation formats to meet the different needs of users [6–10]. In addition, optical signals are susceptible to various transmission impairments that change dynamically in time, which places new demands on optical receivers. At the digital receiver, demodulate the transmitted signal; it is necessary to know the type of modulation format; therefore, correctly

identifying the modulation format is critical for high-quality communication [11–13].

Currently, existing algorithms for modulation format identification (MFI) are broadly classified into two categories according to the identification principle of the algorithm. One is the likelihood-based (LB) method of hypothesis testing based on the likelihood function, which minimizes the probability of false identification and provides the optimal solution on Bayesian ideals [14–16]; however, this method contains higher computational complexity and is not easy to implement. Another category is the feature-based (FB) identification method, which is also the approach taken in this paper. This method has to extract significant features from the received signal and then identify the signal modulation format. Although FB is a sub-optimal method, it is usually more straightforward and easier to implement and can also provide near-optimal identification performance if properly designed [17–20]. For example, an MFI scheme based on lightweight convolutional neural networks (CNN) in the 2D-Stokes planes is proposed in [17]. The authors in [18] proposed an MFI scheme that depends on using CNNs on constellation diagrams. A blind optical MFI scheme based on the Hough transform of the constellation diagram is proposed in [19], which is used to display phase and magnitude information. In [20], the authors proposed a modulation classification scheme based on signal constellation diagram and deep learning to identify different modulated signals. Most of the above MFI techniques can identify modulation formats such as QPSK, 16QAM, and 64QAM. To the authors' knowledge, MFI has not been considered in the literature for higher-order QAM signals.

In this paper, an MFI scheme based on signal constellation diagrams and support vector machine (SVM) is proposed. Firstly, the signal constellation diagrams are divided by the fractal dimension of the weighted linear least squares (WLS-FD) algorithm, and each region's fractal dimension (FD) is calculated, regarded as one of the image features. Then, the feature values of the image in different directions are extracted by the gray-level co-occurrence matrix (GLCM), and their mean and variance are calculated, which is regarded as another feature. Finally, the two features are input into the modulation format classifier constructed by the SVM to achieve MFI in coherent optical communication systems. To verify the feasibility and superiority of the scheme, we compare it with the MFI scheme based on higher-order statistical (HOS) features, GLCM features, and FD features, respectively. Further, we built a 30 GBaud coherent optical communication system with fiber lengths of 80 km and 120 km, where the optical signal-to-noise ratio (OSNR) ranges from 0 dB to 30 dB. The proposed MFI scheme identifies seven modulation formats: QPSK, 8QAM, 16QAM, 32QAM, 64QAM, 128QAM, and 256QAM. The results show that compared with the other three schemes, our proposed scheme has a better identification accuracy at low OSNR. In addition, the identification accuracy of this scheme can reach 100% when the OSNR $\geq$ 10 dB.

## 2. Theory and Principle

The architecture of the MFI scheme proposed in this paper is shown in Figure 1. Firstly, the signal constellation diagrams are divided by the WLS-FD algorithm, and the FD in each region is calculated, which is regarded as one of the image features. Then, the feature values of the image in different directions are extracted by the GLCM, and their mean and variance are calculated, which is regarded as another feature. Finally, the two features are input into the modulation format classifier constructed by the SVM. Meanwhile, to facilitate the learning of feature data by the SVM learning algorithm, it is necessary to change the feature data into one-dimensional data. The training feature data and test feature data are obtained by randomly extracting the feature set several times according to the ratio of 7:3, and the number of samples is 6370 and 2730, respectively.

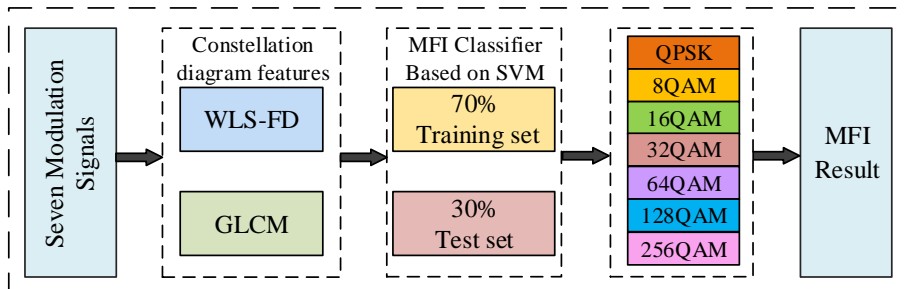

**Figure 1.** The architecture of the MFI scheme.

### 2.1. Extraction of Signal Constellation Diagram Features

The characteristics of the image surface can be described using FD [21,22]. Larger fractal dimension values indicate more complex image surfaces, which usually correspond to rougher textures. In contrast, images with smooth surfaces typically have relatively small fractal dimension values. Based on the above considerations, the image FD is chosen as one of the features to measure image quality. The box-counting (BC) method proposed by Sarkar and Chaudhari is used to calculate the fractal dimension of the images [23]. Although the BC method can estimate the BC of the fractal image, there are certain defects, and it can not reflect the inhomogeneity of this graph, so the differential box-counting (DBC) method is used. Most researchers employed linear least squares (LLS) regression to calculate the FD of the images; however, it is susceptible to outliers. Furthermore, it provides equal weight to all points while drawing a line, which is impracticable; therefore, in this paper, the FD of the images is obtained using WLS regression, and each fitting point is given a distinct weight using the trapezoidal membership function (TMF). As a mature statistical image analysis method [24], GLCM has excellent flexibility and robustness, and the technique is simple, which is extensively utilized in the field of statistical analysis. The co-occurrence matrix is acquired by calculating the grayscale image and its different features in different directions to obtain the feature values. Finally, the mean and variance of these feature values are calculated as the image features.

The algorithm flow for extracting signal constellation diagram features is shown in Figure 2. When calculating the fractal dimension of the constellation diagrams as an input image, because the gray value in the area of the vector point concentration is generally higher than other blank areas, the calculated fractal dimension is bound to have a big difference. The constellation diagrams are divided into 16 * 16 regions, and the subsequent dimensional judgments and fractal box dimensions are calculated for each region separately. WLS-FD sorts the fractal box dimensions calculated from each small region according to the region's location as one of the image features.

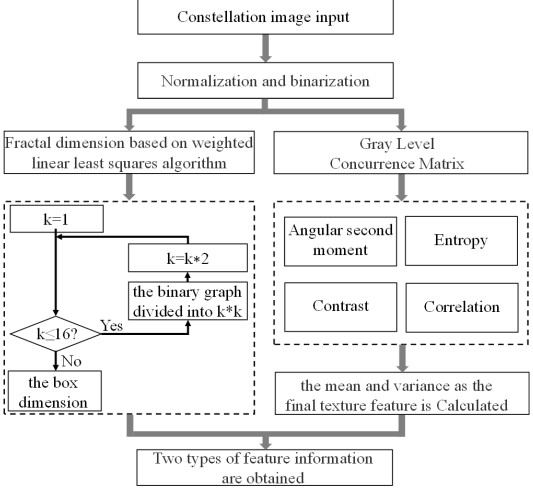

**Figure 2.** Algorithm flow for extracting signal constellation diagram features.

Calculating the constellation diagrams is similar to constructing a three-dimensional surface on which pixels are located along $(x, y)$, and their grayscale values represent the third coordinate $z$. The fractal image of pixel size $M * M$ is divided into small pieces of size $s * s$, where $\frac{M}{2} \geq s > 1$ and $s$ is an integer. The scale of the grid is $r$, which is calculated by $\frac{s}{M}$. Each small block has a column of $s * s * h$ boxes. Where $G$ is the total amount of gray levels in an 8-bit grayscale image, which is 256, and the height $h$ of the box can be formulated as [23] Equation (1):

$$h = \frac{s}{M} \times G \tag{1}$$

The image's highest and lowest gray levels at position $(i, j)$ fall in the $u^{th}$ and $v^{th}$ boxes, respectively, and the total amount of boxes that are size $s * s \times h$ comes to $n_r$, which is obtained by Equation (2):

$$n_r(i, j) = u - v + 1 \tag{2}$$

The value of all grids is $N_r$, which can be obtained by Equation (3):

$$N_r = \sum_{i,j} n_r(i, j) \tag{3}$$

The weight for each grid size is assigned by WLS-FD using the rules of the TMF. Each point $(x_d, y_d)$ on the fitting line corresponds to a different grid size $s$, which is assigned the weight $W(s)$. Where $x_d = \log \frac{1}{r}$, $y_d = \log N_r$, the weight of the point $(x_d, y_d)$ is written as $W(s)$, which can be used with $w_s$ instead of each other. The weight for each data point is assigned by TMF using Equation (4):

$$W(s) = \begin{cases} 0, & s \leq a_t \\ \frac{s - a_t}{b_t - a_t}, & a_t < s < b_t \\ 1, & b_t \leq s \leq c_t \\ \frac{d_t - s}{d_t - c_t}, & c_t < s < d_t \\ 0, & s \geq d_t. \end{cases} \tag{4}$$

According to Equation (4), TMF divides all data points into five parts with intervals of $(-\infty, a_t]$, $(a_t, b_t)$, $[b_t, c_t]$, $(c_t, d_t)$, and $[d_t, +\infty)$, respectively. The points in the first and fifth parts have a weight of 0, while the points in the third part have a weight of 1. The points in the second and fourth parts have weights in the range $(0, 1)$. In this paper, the values of $a_t$, $b_t$, $c_t$, and $d_t$ are estimated by Equation (5):

$$\begin{cases} a_t = s_{min} - 1 \\ b_t = \left| \sqrt[3]{M} \right| \\ c_t = max \left\{ s \left[ \left( \frac{M}{s} \right) + 1 \leq \left( \frac{M}{s - 1} \right) \right] \right\} \\ d_t = s_{max} + 1 \end{cases} \tag{5}$$

Since the size of all the images we use is $512 \times 512$ pixels, we can obtain $a_t = 1$, $b_t = 8$, $c_t = 27$, and $d_t = 257$ by Equation (5). After the weight of the points $(x_d, y_d)$ are obtained. Fitting these points to the regression line $y = ax + b$ via WLS. The FD of the image is calculated by Equation (6):

$$D_A = \frac{\left( \sum_r w_s x_d y_d \right) \left( \sum_r w_s \right) - \left( \sum_r w_s y_d \right) \left( \sum_r w_s x_d \right)}{\left( \sum_r w_S x_d^2 \right) \left( \sum_r w_S \right) - \left( \sum_r w_s x_d \right)^2} \tag{6}$$

The GLCM is an $n * n$ square matrix, where $n$ is the number of gray-level categories in the image. The $(i, j)$ elements in the matrix represent the number of pixel pairs. By calculating the GLCM in four directions $(0°, 45°, 90°$ and $135°)$, $GLCM[p(i, j)]$ is normalized by Equation (7):

$$\frac{p(i,j)}{\varepsilon} \Rightarrow \hat{p}(i,j) \tag{7}$$

where $\varepsilon$ is the constant of normalization.

Angular second moment $m_1$, entropy $m_2$, contrast $m_3$, and correlation $m_4$ are used to perform image block retrieval, which can be calculated by [24] Equation (8):

$$\begin{cases} m_1 = \sum_i \sum_j P(i,j)^2 \\ m_2 = -\sum_i \sum_j P(i,j) \log P(i,j) \\ m_3 = \sum_i \sum_j (i-j)^2 P(i,j) \\ m_4 = \dfrac{\left[\sum_i \sum_j ((i*j)*P(i,j)) - \mu_x \mu_y\right]}{\sigma_x \sigma_y} \end{cases} \tag{8}$$

where: $\mu_x = \sum_i \sum_j i * P(i,j)$, $\mu_y = \sum_i \sum_j j * P(i,j)$, $\sigma_x = \sum_i \sum_j (i-\mu_x)^2 * P(i,j)$, $\sigma_y = \sum_i \sum_j (j-\mu_y)^2 * P(i,j)$. The final texture features in this paper are calculated using the mean and variance of $m_1$, $m_2$, $m_3$, and $m_4$.

*2.2. MFI Classifier Based on SVM*

The SVM [25] is used to build the MFI classifier for processing the extracted signal constellation diagram features. The robustness of the classification algorithm is improved by constructing a linear segmentation hyperplane with maximum boundaries to classify all data correctly. A training dataset $D = \{(x_1, y_1), (x_2, y_2), \cdots, (x_M, y_M)\}$ containing $M$ samples is given. The samples' feature vectors and labels are $x_i$ and $y_i$, respectively, where the feature vector satisfies $x_i = \left(x_i^{(1)}, x_i^{(2)}, \cdots, x_i^{(m)}\right)$ with $m$ features and the labels satisfy $y_i \in \{-1, +1\}$. To train the maximum margin model with parameters $w \in R^D$ and $b \in R$, it solves the following optimization problem:

$$\min_{w,b} \frac{1}{2} \|w\|^2 \tag{9}$$
$$\text{Subject to } y_i\left(w^T x_i + b\right) \geq 1, \; i = 1, \cdots, N$$

According to the feature vector $x_i$ and Lagrange multiplier pairs $\lambda_i$, $\lambda_i^*$ [26], the classification decision function is calculated by Equation (10):

$$f(x) = \sum_{m=1}^{M} (\lambda_i - \lambda_i^*)\langle x_i, x \rangle + b \tag{10}$$

The classifier's performance is measured by comparing the predicted results with $y_i$, and the classification accuracy is expressed by Equation (11):

$$P = \frac{1}{M} \sum_{m=1}^{M} \text{II}(f(x_i) = y_i) \tag{11}$$

where II denotes the indicative function, which is 1 when the condition inside the brackets holds and 0 otherwise. According to Equation (11), the accuracy of the classifier is obtained by adding the number of predicted samples with the same labels as the original ones and dividing them by the total number of samples $M$.

## 3. Simulation Setup and Results

*3.1. Simulation Setup*

The configuration of the simulation system is depicted in Figure 3. First, the pseudo-random bit sequence (PRBS) is generated and used to generate different modulated signals in the arbitrary waveform generator (AWG), where the digital-to-analog converter (DAC)

module turns the digital signal into an analog signal. In the Mach–Zehnder modulator (MZM), four signals are modulated to two optical carriers with different polarization states. The optical carriers are generated by a continuous-wave (CW) laser with a center frequency of 193.1 THz and divided into different polarization states by a polarization beam splitter (PBS). Then, the two polarization states of the modulated optical signal are combined into one using a polarization beam combiner (PBC). The transmission links are 80 km and 120 km single-mode fiber (SMF) with an attenuation coefficient of 0.2 dB/km and a dispersion coefficient is 16.75 ps/nm/km. The amplification gains of the erbium-doped fiber amplifier (EDFA) are 16 dB and 24 dB, respectively. Then, the optical signals are transmitted through SMF and amplified by EDFA, which can completely compensate for the fiber attenuation loss in the process of optical signal transmission. Two optical signals, I and Q, are obtained at the receiver, and the photodiode is used to modulate the optical signals into analog signals. Next, after the coherent receiver, the analog signal is converted into a digital signal by the analog-to-digital converter (ADC) module. Further, dispersion compensation (DC), resampling, timing recovery, and constant mode algorithm (CMA) equalization are performed sequentially at the DSP. Finally, the required constellation diagrams are generated. The parameters of the coherent optical communication system are shown in Table 1.

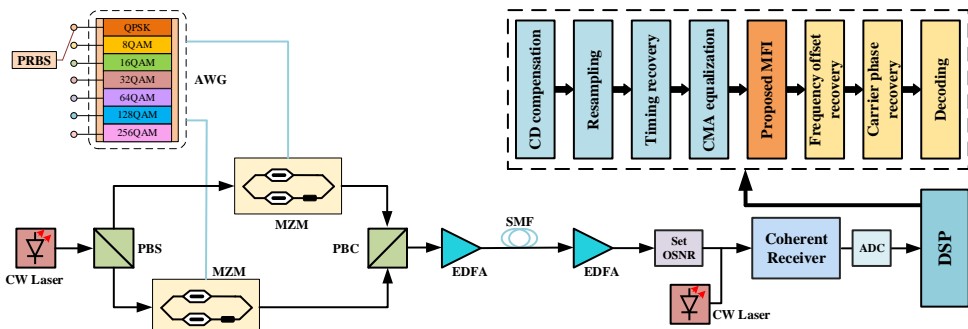

**Figure 3.** Simulation system setup.

**Table 1.** Cohert optical fiber communication system parameters.

| Parameter Type | Value |
|---|---|
| Central wavelength | 193.1 THZ |
| Signal Rate | 30 GBaud |
| Fiber input power | 10 dBm |
| Optical amplifier | EDFA |
| EDFA gain | 16 dB, 24 dB |
| EDFA noise | 4 dB |
| Fiber | SMF |
| Fiber attenuation coefficient | 0.2 dB/km |
| Fiber nonlinear coefficient | 1.31 $(\text{W·km})^{-1}$ |
| Fiber dispersion coefficient | 16.75 ps/(nm·km) |
| Fiber distance | 80 km, 120 km |
| OSNR range | 0 dB∼30 dB |

### 3.2. Results and Discussion

The simulation system generated 9100 constellation diagrams in PNG format, which have a pixel size of 512 × 512. The sample constellation diagram obtained after 80 km SMF transmission is shown in Figure 4.

| OSNR | 0 dB | 2.5 dB | 5 dB | 7.5 dB | 10 dB | 12.5 dB | 15 dB | 17.5 dB | 20 dB | 22.5 dB | 25 dB | 27.5 dB | 30 dB |
|---|---|---|---|---|---|---|---|---|---|---|---|---|---|
| QPSK | | | | | | | | | | | | | |
| 8QAM | | | | | | | | | | | | | |
| 16QAM | | | | | | | | | | | | | |
| 32QAM | | | | | | | | | | | | | |
| 64QAM | | | | | | | | | | | | | |
| 128QAM | | | | | | | | | | | | | |
| 256QAM | | | | | | | | | | | | | |

| OSNR | 0dB | 2.5dB | 5dB | 7.5dB | 10dB | 12.5dB | 15dB | 17.5dB | 20dB | 22.5dB | 25dB | 27.5dB | 30dB |
|---|---|---|---|---|---|---|---|---|---|---|---|---|---|
| QPSK | | | | | | | | | | | | | |
| 8QAM | | | | | | | | | | | | | |
| 16QAM | | | | | | | | | | | | | |
| 32QAM | | | | | | | | | | | | | |
| 64QAM | | | | | | | | | | | | | |
| 128QAM | | | | | | | | | | | | | |
| 256QAM | | | | | | | | | | | | | |

**Figure 4.** The sample constellation diagram obtained after 80 km SMF transmission.

Seven modulation signal samples are used in the test set for the simulation test, according to the proposed MFI scheme. As shown in Figures 5 and 6, the overall and average identification accuracy at low OSNR are plotted in the 3D histograms for the seven signals transmitted through 80 km and 120 km fiber, respectively.

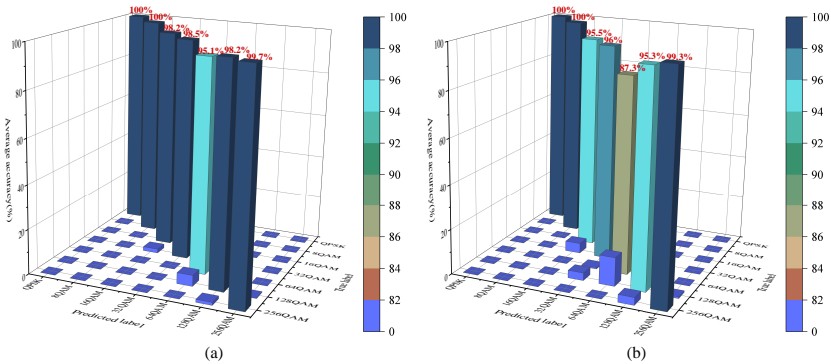

**Figure 5.** The average identification accuracy after 80 km fiber transmission (**a**) OSNR range is 0∼30 dB (**b**) OSNR range is 0∼10 dB.

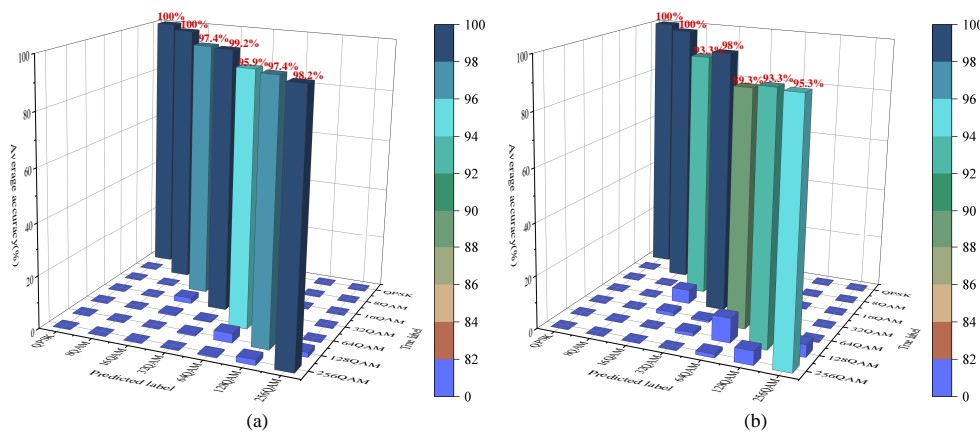

**Figure 6.** The average identification accuracy after 120 km fiber transmission (**a**) OSNR range is 0~30 dB (**b**) OSNR range is 0~10 dB.

In Figure 5, the three types of signals, QPSK, 8QAM, and 256QAM, have high identification accuracy of 99% in the overall case and at low OSNR, with the identification accuracy of QPSK and 8QAM signals reaching 100%. The 16QAM, 32QAM, 64QAM, and 128QAM signals have the possibility of misidentification among each other. However, in the MFI scheme proposed in this paper, the overall identification accuracy of 16QAM, 32QAM, 64QAM, and 128QAM reaches 98.2%, 98.5%, 95.1%, and 98.2%, respectively, and the identification accuracy at low OSNR reaches 95.5%, 96%, 87.3%, and 95.3%, respectively.

In Figure 6, the three types of signals, QPSK, 8QAM, and 32QAM, have high identification accuracy of 98% in the overall case and at low OSNR, with the identification accuracy of QPSK and 8QAM signals reaching 100%. The 16QAM, 64QAM, 128QAM, and 256QAM signals have the possibility of misidentification among each other. However, in the MFI scheme proposed in this paper, the overall identification accuracy of 16QAM, 64QAM, 128QAM, and 256QAM reaches 97.4%, 95.9%, 97.4%, and 98.2%, respectively, and the identification accuracy at low OSNR reaches 93.3%, 89.3%, 93.3%, and 95.3%, respectively.

In Figure 7, the identification accuracy of the seven types of signals gradually increases as the OSNR increases until they all reach 100%. Among them, the identification accuracy of QPSK and 8QAM is always 100%. In Figure 7a, the OSNR values of 16QAM, 32QAM, 64QAM, 128QAM, and 256QAM when reaching 100% identification accuracy are around 5 dB, 2.5 dB, 7.5 dB, 10 dB, and 2.5 dB, respectively. In Figure 7b, the OSNR values of 16QAM, 32QAM, 64QAM, 128QAM, and 256QAM at 100% identification accuracy are around 5 dB, 2.5 dB, 5 dB, 10 dB, and 7.5 dB, respectively. The above analysis shows that the proposed MFI scheme can significantly improve the signal identification accuracy, thus improving the classifier's performance.

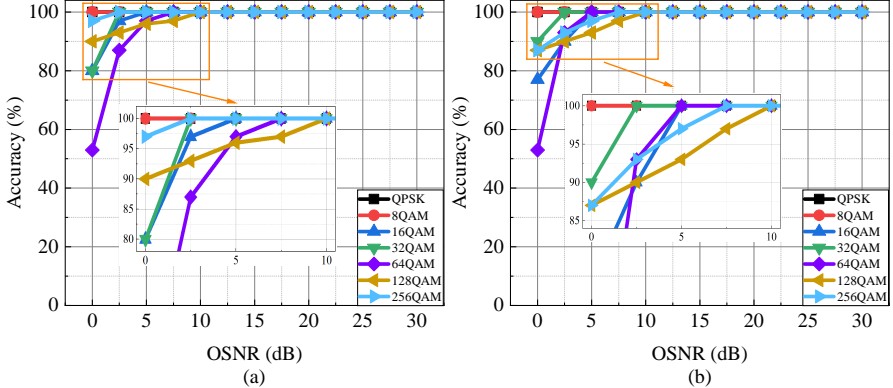

**Figure 7.** Comparison of seven types of signal identification accuracy. The fiber distance is set as (**a**) 80 km, (**b**) 120 km.

To verify the superiority of the proposed MFI scheme in this paper, different MFI schemes are compared at fiber lengths of (a) 80 km and (b) 120 km, as depicted in Figure 8. The results show that the proposed MFI scheme quickly achieves 100% overall identification accuracy and remains stable when OSNR $\geq$ 10 dB. High overall identification accuracy is also achieved at low OSNR. Because of the influence of high-order QAM signals in seven kinds of signals, such as 64, 128, and 256QAM, the overall identification accuracy of GLCM features and FD features is not good. Although the overall identification accuracy of MFI based on HOS features is high, it fails to achieve 100% identification accuracy within the set OSNR. In summary, the proposed MFI scheme has better stability and superiority compared with other MFI schemes.

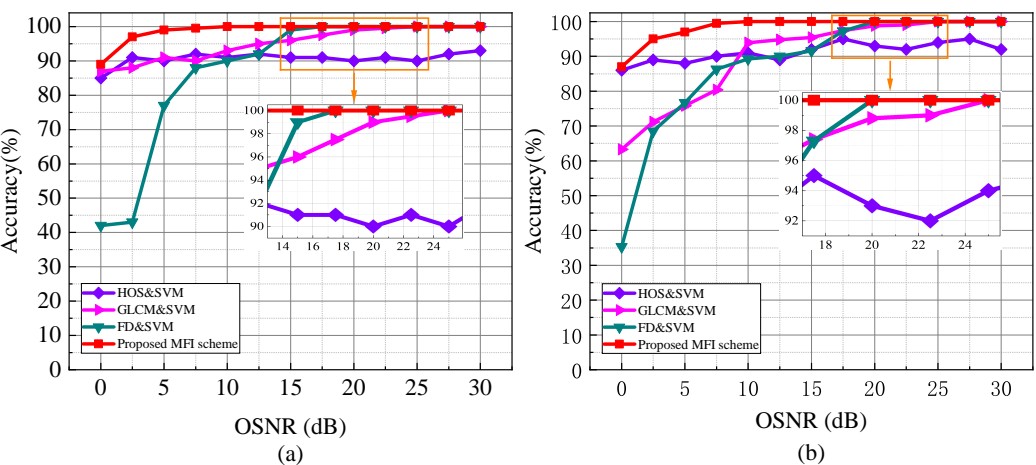

**Figure 8.** Comparison of the overall identification accuracy of the seven signals with different MFI schemes. The fiber distance is set as (**a**) 80 km, (**b**) 120 km.

Finally, to understand the computational complexity of the proposed algorithm, we calculated the CPU running time required to extract the features of a single constellation diagram and compared it with the other three algorithms, as shown in Figure 9. CPU running time tests were conducted on an Intel Personal Computer with Processor Core i5-10210U CPU at 1.60 GHz, 16 GB Random Access Memory, and Windows 10 Home Edition operating system. Combined with Figure 8, it can be seen that the proposed MFI scheme achieves a significant improvement in identification accuracy and stability at the cost of minor computational complexity and is the result of balancing accuracy and stability optimally.

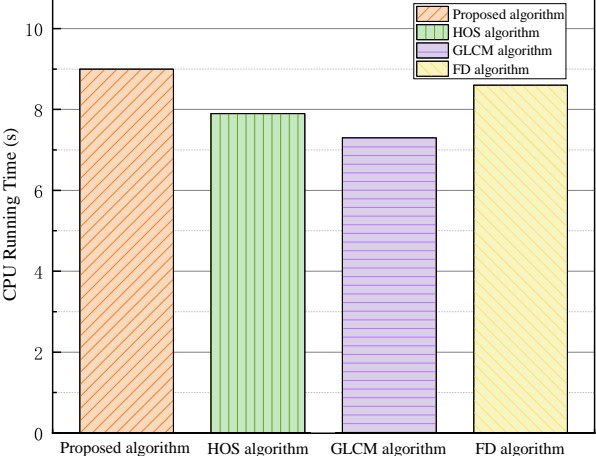

**Figure 9.** Comparison of CPU running time of different algorithms for extracting constellation diagram features.

## 4. Conclusions

In this paper, an MFI scheme based on signal constellation diagrams and SVM is proposed. Firstly, the signal constellation diagrams are divided by the WLS-FD algorithm, and each region's FD is calculated, regarded as one of the image features. Then, the feature values of the image in different directions are extracted by the GLCM, and their mean and variance are calculated, which is regarded as another feature. Finally, the two features are input into the modulation format classifier constructed by the SVM to achieve MFI in coherent optical communication systems. To verify the feasibility and superiority of the scheme, a 30 GBaud coherent optical communication system with fiber lengths of 80 km and 120 km is constructed. The results show that compared with other traditional schemes, the proposed MFI scheme greatly improves the identification accuracy with the same OSNR.

**Author Contributions:** Conceptualization, Q.Z.; methodology, H.Y. and L.Y.; software, Z.H. and J.J.; validation, Z.H., Q.Z., and R.G.; formal analysis, Z.H., F.T., and B.L.; investigation, Z.H.; resources, F.W. and Q.T.; data curation, Z.H. and Y.W.; writing—original draft preparation, Z.H.; writing—review and editing, Z.H., Q.Z., and H.Y.; visualization, Q.Z. and J.J.; supervision, X.X. and R.G.; project administration, Q.Z. and X.X.; funding acquisition, Q.Z. All authors have read and agreed to the published version of the manuscript.

**Funding:** This work was supported in part by the National Key R&D Program of China under grant number 2019YFB1803701.

**Institutional Review Board Statement:** Not applicable.

**Informed Consent Statement:** Not applicable.

**Data Availability Statement:** Data underlying the results presented in this paper are not publicly available at this time but may be obtained from the authors upon reasonable request.

**Conflicts of Interest:** The authors declare no conflict of interest.

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
