# Peer review of "Modulation Format Identification Based on Signal Constellation Diagrams and Support Vector Machine"

_photonics, doi:10.3390/photonics9120927_

Round 1
Reviewer 1 Report
This paper proposed a modulation format identification scheme based on a signal constellation diagram and support vector machine. The following comments are listed for the authors to improve the manuscript before publication further.
1. In Table 1, how is the gain of EDFA calculated?
2. In Figure 5 and Figure 6, the visual judgment of Average accuracy size is unclear, and it would be better to label the value in the figure.
3. In section 3.2, the authors stated, "In summary, the proposed MFI scheme has better stability and superiority than other MFI schemes.”. Can you explain how stability and superiority are manifested, respectively?
Reviewer 2 Report
The authors proposed a feature-based MFI algorithm for high-order QAM-encoded coherent optical transmission systems. The principle of the proposed algorithm was described in detail and the numerical simulation was also performed to verify its performance. Besides, performance comparison was also done with the other three algorithms and the relevant results exhibited that the proposed MFI algorithm can provide better identification accuracy at low OSNRs. The reviewer recommends accepting this manuscript after minor revisions. My comments on this manuscript are given as follows.
1) It is suggested to analyze the computational complexity of the proposed MFI for comparison with the other three MFI algorithms.
2) Inappropriate descriptions and typos should be corrected. For example, “the binary signals at the transmitter side are modulated by quadrature modulation and digital-to-analog conversion (DAC) to the electrical signal.”, “Mach-Zehnder multiplex (MZM)”, “the analog signals are converted into digital signals by the analog-to-digital converter (ADC) module after damage compensation in the offline DSP module”, etc.
3) Figure 3 makes me very confused. What’s the AWG in the transmitter? Where are the MZM, DAC and ADC in Fig. 3, as described in Section 3.1? What’s the module after the ECL and before EDFA? Are two IQ modulators employed for PDM? Please redraw this figure and add appropriate explanations.
4) In Figs. 4-7, the low identification accuracy for 64QAM was observed with the proposed MFI algorithms. Please explain it in detail.
5) Please specify the modulation format for the results presented in Fig. 8.
